# Highly Porous SnO₂/TiO₂ Heterojunction Thin-Film Photocatalyst Using Gas-Flow Thermal Evaporation and Atomic Layer Deposition

Sungjin Kim [1,2], Hyeon-Kyung Chang [1,2], Kwang Bok Kim [3], Hyun-Jong Kim [1], Ho-Nyun Lee [1], Tae Joo Park [2] and Young Min Park [1,*]

[1] Heat and Surface Technology R&D Department, Korea Institute of Industrial Technology (KITECH), Incheon 21999, Korea; sungjin@kitech.re.kr (S.K.); hkjjj0902@gmail.com (H.-K.C.); hjkim23@kitech.re.kr (H.-J.K.); hnlee@kitech.re.kr (H.-N.L.)

[2] Department of Materials Science and Chemical Engineering, Hanyang University, Ansan 15588, Korea; tjp@hanyang.ac.kr

[3] Digital Health Care R&D Department, Korea Institute of Industrial Technology (KITECH), Cheonan 31056, Korea; kb815kim@kitech.re.kr

* Correspondence: youngmin@kitech.re.kr; Tel.: +82-10-4278-7958

**Abstract:** Highly porous heterojunction films of SnO₂/TiO₂ were prepared using gas-flow thermal evaporation followed by atomic layer deposition (ALD). Highly porous SnO₂ was fabricated by introducing an inert gas, Ar, during thermal evaporation. To build heterogeneous structures, the TiO₂ layers were conformally deposited on porous SnO₂ with a range of 10 to 100 cycles by means of ALD. The photocatalytic properties for different TiO₂ thicknesses on the porous SnO₂ were compared using the degradation of methylene blue (MB) under UV irradiation. The comparisons showed that the SnO₂/TiO₂-50 heterostructures had the highest photocatalytic efficiency. It removed 99% of the MB concentration, and the decomposition rate constant (K) was 0.013 min$^{-1}$, which was approximately ten times that of the porous SnO₂. On the other hand, SnO₂/TiO₂-100 exhibited a lower photocatalytic efficiency despite having a TiO₂ layer thicker than SnO₂/TiO₂-50. After 100 cycles of TiO₂ ALD deposition, the structure was transferred from the heterojunction to the core–sell structure covered with TiO₂ on the porous SnO₂, which was confirmed by TEM analysis. Since the electrons photogenerated by light irradiation were separated into SnO₂ and produced reactive oxygen, O₂$^-$, the heterojunction structure, in which SnO₂ was exposed to the surface, contributed to the high performance of the photocatalyst.

**Keywords:** photocatalyst; heterojunction; thermal evaporation deposition; atomic layer deposition; porous tin dioxide; titanium dioxide; core-shell structure

## 1. Introduction

Water contamination with dye discharges from various industries is considered a significant threat to the environment and public health [1,2]. Moreover, the scarcity of water resources has also become a major problem. Additionally, conventional wastewater treatment technologies have been problematic because of their long operation times and high costs [3,4]. Among the various dye-contaminated water treatment methods, photocatalytic degradation has attracted much attention for the removal of organic residues from wastewater. In particular, metal oxide heterojunctions employing advanced oxidation processes (AOPs) have been widely explored because of their rapid purification time and environmentally benign processes [5,6]. Under light irradiation, the heterojunction of a metal oxide semiconductor facilitates the separation of the generated electron–hole (e$^-$/h$^+$) pairs. These generated excitons produce reactive oxygen species in aqueous media, such as hydroxyl radicals (·OH) and superoxides (·O₂$^-$), which are known to be photocatalytically active for the decomposition of organic contaminants, thereby purifying

dye-contaminated water [7–9]. Among various metal oxides, $TiO_2$ has been considered an attractive photocatalyst material for dye degradation because of its excellent catalytic activity and high chemical stability [10–15]. Under UV irradiation, the electrons generated by light absorption form a radical on the surface of $TiO_2$ near the energy level of the valence band, thus removing the dye from the water. However, the large band gap, ~3.2 eV, and fast recombination of the exciton of $TiO_2$ limit the photocatalytic efficiency. To resolve the low light adsorption and fast recombination of excitons in a single metal oxide, the heterostructures of $TiO_2$ with other metal oxide semiconductors, such as ZnO, and $SnO_2$, etc., have been explored using various synthesis methods. Metal oxide semiconductors with lower valence bands than $TiO_2$, such as $SnO_2$, facilitate the transfer of photogenerated holes in the valence band to $TiO_2$ in the heterojunction, thus hindering charge recombination and leading to efficient photocatalysis [16,17]. In addition to the selection of appropriate materials in heterojunctions, most research efforts in the field of photocatalysts have focused on optimizing microstructures to increase light absorption efficiency and to enlarge the active surface area using particle-type metal oxides. Since nanoparticle-type photocatalysts enlarge the active surface area and enhance light absorption, the synthesis of metal oxide nanoparticles has been widely investigated as a high-performance photocatalyst. However, despite their high performance, particle-type photocatalysts require additional separation processes in the suspension system, leading to high costs. Additionally, they cause secondary contamination resulting from the organic and inorganic binders in the synthesis and deposition process, thereby limiting their application [18–20]. Furthermore, most fabrication methods for particle-type heterojunctions involve complicated preparation processes and exhibit lower microstructure and chemical composition reproducibility [21]. Therefore, the development of a film-type photocatalyst with reproducibility is necessary to extend the usability and to improve the efficiency of the photocatalyst.

In this study, we fabricated a nanoporous $SnO_2/TiO_2$ heterojunction on a planar substrate using gas-flow thermal evaporation followed by atomic layer deposition (ALD) without binders. The flow of an inert gas, Ar, with a pressure of 0.2 Torr during thermal evaporation, allows the formation of a nanoporous structure. The conformal growth of $TiO_2$ on porous $SnO_2$ was demonstrated with a different cycle of ALD. In order to not cover the entire surface of $SnO_2$, $TiO_2$ was deposited with a lower number of ALD cycles, from 10 to 100 cycles, corresponding to a thickness of less than 2 nm. Chemical and microstructural analyses revealed the formation of the $SnO_2/TiO_2$ heterostructure. The evaluation of the photocatalytic activities of the as-prepared highly porous $SnO_2/TiO_2$ structures used the degradation of a methylene blue (MB) dye solution with a low UV irradiation intensity. Therefore, the results reveal that the heterostructure, not the core-shell structure of $SnO_2/TiO_2$, induces high photocatalytic activity, improves charge separation, and utilizes separated electrons and holes for photocatalysis in $TiO_2$ and $SnO_2$, respectively.

## 2. Results and Discussion

The highly porous heterojunction photocatalyst $SnO_2/TiO_2$ was fabricated by gas-flow-modified thermal evaporation followed by ALD. The plan-view and cross-sectional SEM images, Figure 1a,b, respectively, clearly show that the 3D nanoform microstructure of $SnO_2$ is well deposited with gas-flow-modified thermal evaporation at a pressure of 0.2 Torr. The surface area of the porous $SnO_2$ and $SnO_2/TiO_2$ was measured as about 74 $m^2/g$ and 72.5 $m^2/g$ with the Brunauer–Emmett–Teller method, respectively. The 50 cycles of ALD on the $SnO_2$ also did not significantly change the porosity of the $SnO_2$, as shown in Figure 1c,d, since the corresponding thickness by the ALD process is about 3 nm. On the other hand, the EDX elemental mapping on the $SnO_2/TiO_2$-50 clearly shows that the Ti atom was deeply distributed inside the matrix of the $SnO_2$ porous structure, indicating that the $TiO_2$ layer was conformally deposited on the porous nanostructure, as shown in Figure S1. In addition, transmission electron microscopy (TEM) analysis with electron energy loss spectroscopy showed that the $TiO_2$ layers were well deposited on the surface of the porous $SnO_2$ by ALD (Figure 2). As shown in Figure 2a, the nanoform of

$SnO_2/TiO_2$-100 shows that the $TiO_2$ layer is coated on the $SnO_2$ structure, with a thickness of approximately 1.3 nm. The growth rate of the $TiO_2$ on the porous $SnO_2$ structure is as low as one-third of that on the planar substrate because more precursors are required to cover the large surface area of the porous structure (Figure S2). Moreover, the electron energy loss spectroscopy spectrum showed that the $TiO_2$ was uniformly decorated on the whole surface of $SnO_2$, indicating that the $TiO_2$ deposition in 100 ALD cycles could fabricate a core-shell structure (Figure 2b–d. On the other hand, after ALD with less than 50 cycles, the particular structure of $SnO_2$ was not entirely covered by $TiO_2$ (Figure S3). The effect of microstructural changes with the number of ALD cycles on the photocatalytic performance will be further discussed in detail with the results of MB decomposition. The EDXRF analysis also showed that the Ti density increased with the number of ALD cycles, implying that the gradual growth of the $TiO_2$ layer with the number of ALD cycles was achieved on the porous structure (Figure 3). Note that the Ti density of sample after 100 cycles of ALD dramatically increases compared to other samples. Since the porous structure requires more cycles to move from the nucleation stage to the growth stage of ALD due to the larger surface area, it is expected that the $SnO_2$ nanoforms deposited through less than 50 cycles of ALD exhibit a different growth rate compared to $SnO_2/TiO_2$-100. The ALD as-synthesized $TiO_2$ layer was analyzed as the anatase phase using XRD, corresponding to JCPDS no. 21-1272 (Figure S4).

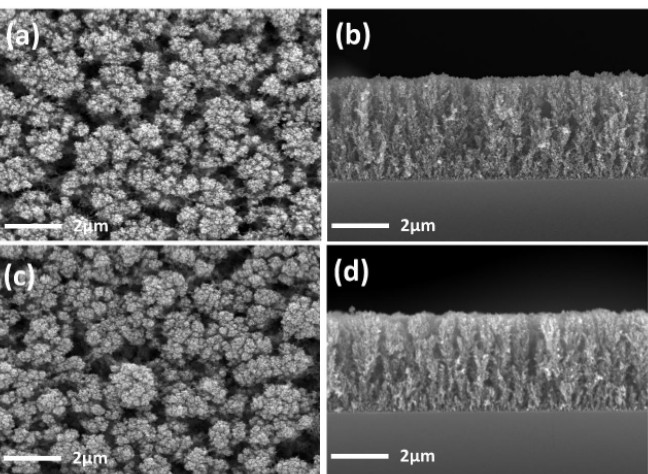

**Figure 1.** FE-SEM analysis of the porous $SnO_2$ and $SnO_2/TiO_2$-50. The surface section of (**a**) $SnO_2$ and (**c**) $SnO_2/TiO_2$-50 and the cross section of (**b**) $SnO_2$ and (**d**) $SnO_2/TiO_2$-50.

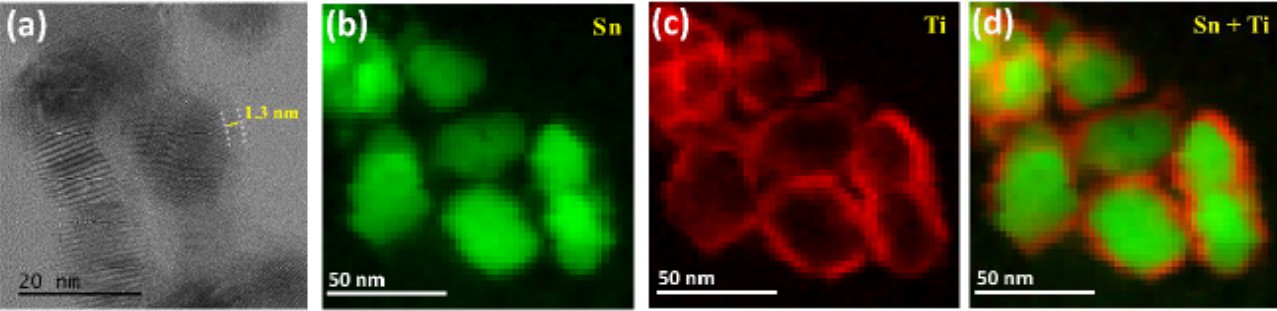

**Figure 2.** (**a**) The TEM image of $SnO_2/TiO_2$-100 and the EELS elemental mapping showing the elemental distribution of (**b**) Sn, (**c**) Ti, and (**d**) their overlay.

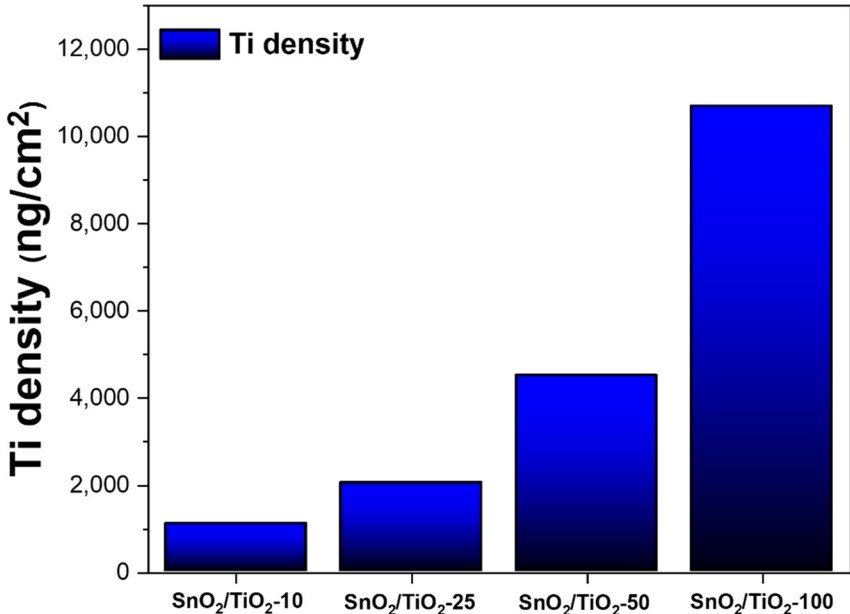

**Figure 3.** The Ti density of the $SnO_2/TiO_2$ photocatalysts analyzed using EDXRF. The Ti density increased with the number of $TiO_2$ ALD cycles.

The elemental chemical states of the constituent element, $SnO_2/TiO_2$, were determined using XPS for the different numbers of ALD cycles. As shown in Figure 4a, the two peaks of the Sn 3d spectra, 486.3 eV and 494.7 eV, assigned as Sn $3d_{5/2}$ and Sn $3d_{3/2}$, respectively, were observed in the spectra of the $SnO_2/TiO_2$-50 nanocomposite, implying that the porous SnO nanostructure was converted to the $SnO_2$ morphology after the annealing process. The two peaks of 458.4 eV and 464.1 eV corresponding to the peaks of Ti $2p_{3/2}$ and Ti $2p_{1/2}$ indicated successful $TiO_2$ deposition for the typical $Ti^{4+}$-O bond on the porous $SnO_2$ matrix, which is in good agreement with the EDXRF and EELS results (Figure 4b). As the amount of $TiO_2$ on the surface of the porous $SnO_2$ increased with the number of ALD cycles, a gradual shift of the Sn 3d peak toward a lower binding energy was observed, which was attributed to the donation of electrons to $TiO_2$ via the formation of Sn-O-Ti at the interface (Figure 4c,d) [22–24]. The introduction of $TiO_2$ on the surface of the $SnO_2$ induced an electron-rich environment at the interface, thereby lowering the binding energy of the $SnO_2$ 3d electrons [25]. As shown in Figure 5, the XPS spectra for O 1s of pristine $SnO_2$ can be deconvoluted into two peaks with binding energies of 530.4 eV and 531.2 eV for the lattice oxygen (Lo) and oxygen vacancy or defect (Vo) of $SnO_2$, respectively [26]. The O 1s binding energy (530.4 eV) of $SnO_2/TiO_2$ gradually decreased with number of $TiO_2$ ALD cycles (Figure 5). Since $TiO_2$ has lower binding energy (529.8 eV) in the oxygen state than $SnO_2$, the formation of a $TiO_2/SnO_2$ heterojunction led to the formation of Sn-O-Ti bonding, thus exhibiting the lower binding energy of the Lo State. Hence, the Lo binding energy was substantially decreased in the $SnO_2/TiO_2$ heterostructure. The Vo state, implying an oxygen vacancy, on the other hand, slightly decreased with increasing $TiO_2$ layers, indicating that the deposition of $TiO_2$ reduced the oxygen vacancies and defects on the $SnO_2$ nanoform. Therefore, XPS analysis revealed that the ALD process not only induced the strongly bound heterojunction of $SnO_2/TiO_2$ but also reduced the surface defects of $SnO_2$, leading to efficient exciton separation and charge transfer.

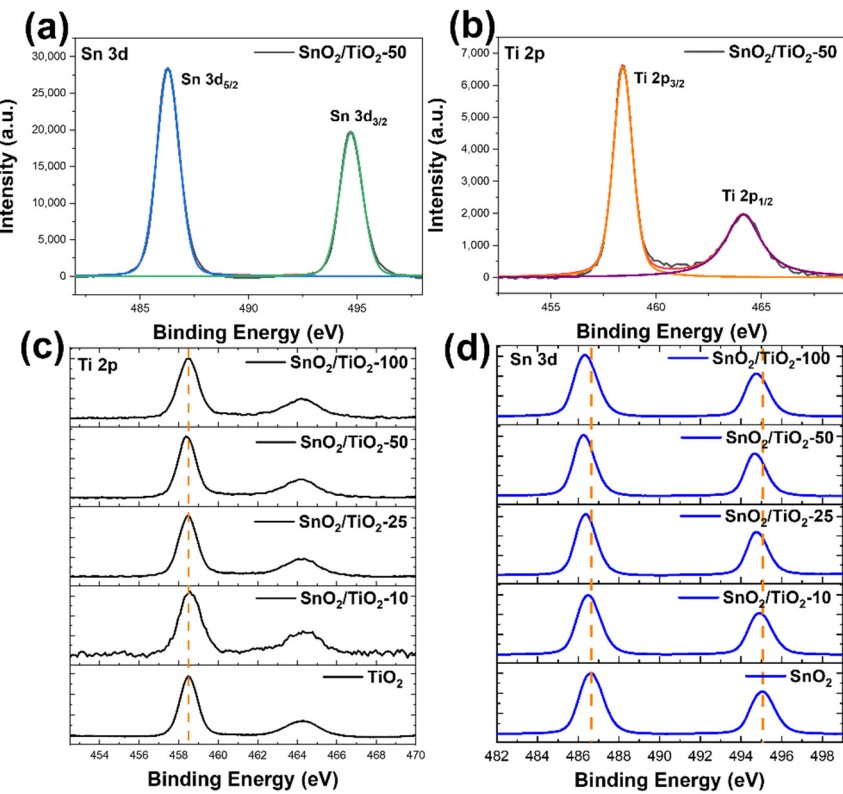

**Figure 4.** The XPS spectrum of the (**a**) Sn 3d and (**b**) Ti 2p states of the $SnO_2/TiO_2$-50 photocatalyst. The peak shift of (**c**) Ti 2p and (**d**) Sn 3d with the increase of $TiO_2$ ALD cycles.

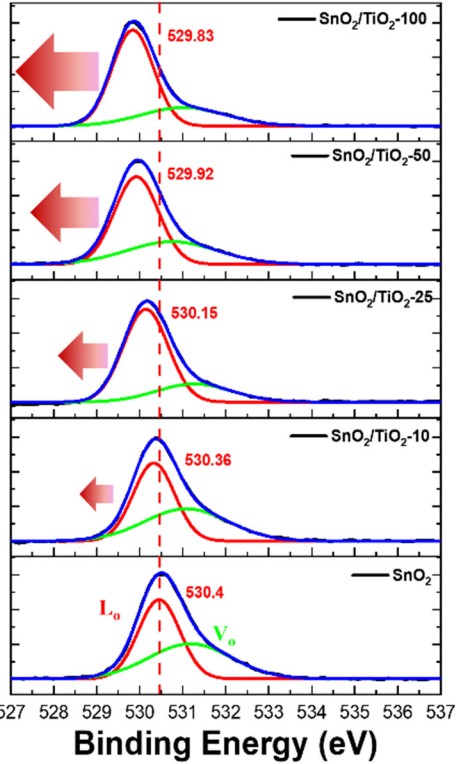

**Figure 5.** Comparison of the O 1s state of the $SnO_2/TiO_2$ photocatalysts with different $TiO_2$ ALD cycles from 10 to 100 cycles. The blue line was the chemical state of O 1s spectra and deconvoluted of lattice oxygen (Lo) and oxygen vacancy (Vo) which was indicated the red and green line, respectively.

The light absorption and band gap with different $TiO_2$ ALD cycles were measured using UV-vis absorption spectra, as shown in Figure 6. The $SnO_2/TiO_2$ samples showed strong absorption intensity at 320 nm with the increasing thickness of the $TiO_2$ nanoscale layers [27]. The nanoform of the porous $SnO_2/TiO_2$ induces more light absorption owing to the greater light pathway of the nanoporous structure. Additionally, the formation of a semiconducting heterojunction structure with $TiO_2$ enhances the absorption intensity at approximately 400 nm. For better understanding, the optical band gaps of the $SnO_2/TiO_2$ samples were experimentally determined using the linear portion of the Kubelka–Munk function. The measured optical band gap of the porous $SnO_2$ nanoform was 3.2 eV whereas all the $SnO_2/TiO_2$ photocatalysts had lower band gaps. Further, the $SnO_2/TiO_2$-10, 25, 50, and 100 samples were slightly shifted toward shorter band gap, with the increasing $TiO_2$ layer at 3.03, 2.93, 2.88, and 3.00 eV, respectively. However, the $SnO_2/TiO_2$-100 samples displayed a wavelength shift toward a larger band gap, which indicated that the thick layer of $TiO_2$ caused less light scattering in the pores and decreased light reflection pathways [28,29].

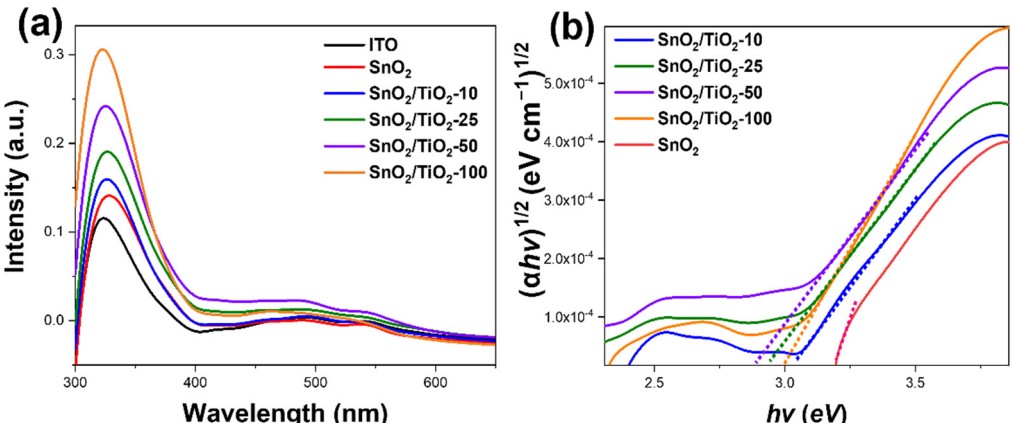

**Figure 6.** (**a**) UV-vis absorption spectra of the porous $SnO_2$ and all the $SnO_2/TiO_2$ photocatalysts on the ITO substrate. (**b**) Calculation of the optical bandgap from the UV-vis absorption spectra of the porous $SnO_2$ nanofoam and all of the the $SnO_2/TiO_2$ photocatalysts.

The photocatalytic performance of the highly porous $SnO_2/TiO_2$-10, 25, 50, and 100 heterojunction films was evaluated using the degradation of the MB solution under UV light irradiation (Figure 7). While the MB solution was barely degraded without the catalysts, the porous $SnO_2$ nanofoam exhibited a 60% MB degradation after 300 min, and the presence of all of the porous $SnO_2/TiO_2$ nanofoams enhanced photocatalytic activity compared to the bare $SnO_2$ nanofoam due to the synergistic effect between the $SnO_2$ and $TiO_2$ heterojunction. The time-dependent UV-vis absorbance spectra of MB degradation displayed a significant reduction in the MB concentration, as shown in Figure S5. The photocatalytic efficiency (η)% of the photocatalyst was calculated from $η = (C_0 − C)/C × 100$, where $C_0$ is the initial concentration of MB, and C is the remaining concentration of MB after photocatalysis. As confirmed in Figure 6 and Figure S6, the $SnO_2/TiO_2$-50 heterostructure photocatalyst displays the highest photocatalytic efficiency (99%) compared to $SnO_2/TiO_2$-10 (73%), $SnO_2/TiO_2$-25 (91%), $SnO_2/TiO_2$-100 (81%), and $SnO_2$ (60%) after photocatalysis. Comparing the photocatalytic reaction rate, which follows pseudo first order reaction kinetics and is expressed as $ln(C_0/C) = −Kt$, where K is apparent rate constant, and t is the time interval, the decomposition rate constant K values were 0.0030, 0.0042, 0.0078, 0.013, and 0.0056 $min^{−1}$ for the $SnO_2$, $SnO_2/TiO_2$-10, 25, 50, and 100 photocatalysts, respectively (Table 1). The rate constant of $SnO_2/TiO_2$-50 was approximately two times higher than that of the core-shell $SnO_2/TiO_2$-100 and had the highest reaction rate compared to the as-fabricated photocatalysts. It is generally acknowledged that photoexcited electrons and holes play a key role in photocatalytic reduction reactions, as they generate lots of

main reactive species that are involved in photocatalytic oxidation, such as $\cdot O_2^-$ and $\cdot OH$. The low photocatalytic efficiency of the porous $SnO_2$ films is attributed to conduction band of the $SnO_2$, which has lower electric potential compared to the $O_2/O_2^-$ oxidation potential, although the holes in valence band of $SnO_2$ oxidized the organic pollutant directly using the $\cdot OH$ radical to contribute to the performance [30]. On the contrary, the $SnO_2/TiO_2$ heterojunction gives rise to charge separation due to the potential difference, hence increasing the lifetime of the charge carrier and improving the interfacial charge transfer to the adsorbed surface. The photogenerated electrons and holes of the $SnO_2/TiO_2$ heterojunction produced strong oxidizing radicals of $\cdot O_2^-$ and $\cdot OH$ from the water or oxygen on the surface of $SnO_2$ and $TiO_2$, respectively, thereby degrading the organic pollutant (Figure 8) [31,32]. Since the $\cdot OH$ radical generated in the valence band of $TiO_2$ is a stronger oxidant, the introduction of the $TiO_2$ layer can improve the photocatalytic efficiency. In addition, light absorption was enhanced by increasing the thickness of the $TiO_2$. Therefore, more ALD cycles of $TiO_2$ induce more active sites for $TiO_2$ and light absorption, thereby improving the photocatalytic performance, as shown in Figure 6. Note that the $SnO_2/TiO_2$-100 photocatalyst had a lower MB degradation performance than the $SnO_2/TiO_2$-50 despite the thicker $TiO_2$ layers and greater light absorption. As shown in Figure 2, 100 cycles of the ALD process produced a core-shell structure and removed the surface of $SnO_2$, which was used to generate $\cdot O_2^-$ oxidizing radicals from the transfer of photogenerated electrons. Furthermore, the thicker $TiO_2$ layer of $SnO_2/TiO_2$-100 hinders the charge transfer to the surface of the $TiO_2$ layer to degrade MB. Therefore, the $SnO_2/TiO_2$-50 heterojunction structure possessing both the $TiO_2$ and $SnO_2$ surfaces is advantageous for efficient photocatalysis since it has enhanced light absorption and more oxidizing radicals from both the photogenerated electrons and holes.

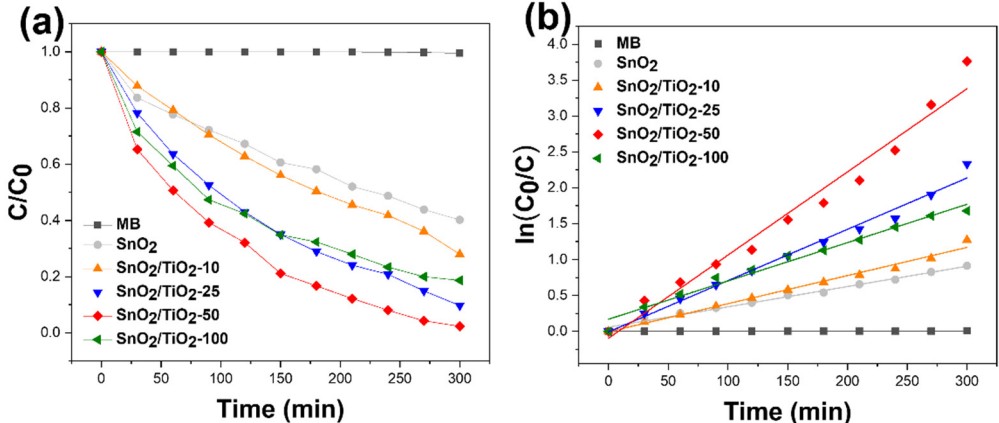

**Figure 7.** (**a**) Decomposition of the MB concentration ratio ($C/C_0$) and (**b**) the kinetics of the photocatalytic degradation of MB by all the $SnO_2/TiO_2$ photocatalysts with different $TiO_2$ ALD cycles.

**Table 1.** Comparison of the photocatalytic efficiency (%) and decomposition reaction rate (*K*) of the porous $SnO_2$ nanofoam and $SnO_2/TiO_2$-10, 25, 50, and 100 heterojunction films.

|  | MB | $SnO_2$ | $SnO_2/TiO_2$-10cyc | $SnO_2/TiO_2$-25cyc | $SnO_2/TiO_2$-50cyc | $SnO_2/TiO_2$-100cyc |
|---|---|---|---|---|---|---|
| **Photo-catalytic efficiency (%)** | <1 | 60 | 73 | 91 | 99 | 81 |
| **K (min$^{-1}$)** | N/A | 0.0030 | 0.0042 | 0.0078 | 0.013 | 0.0056 |

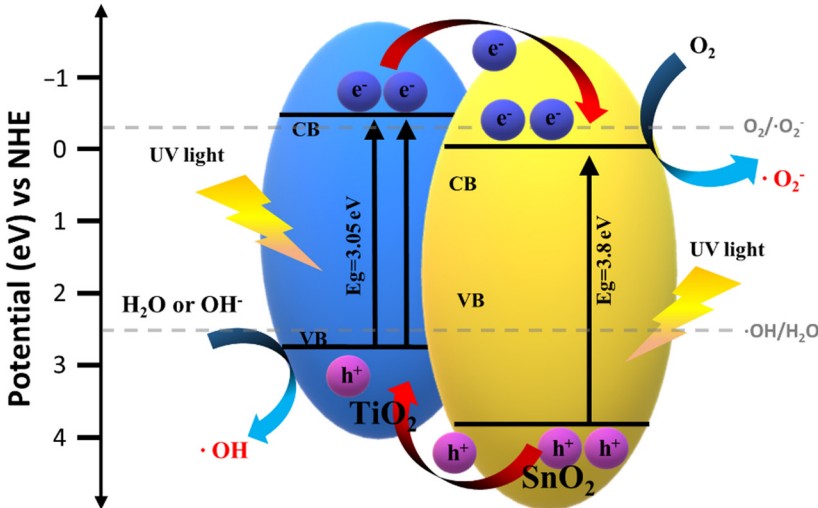

**Figure 8.** Band diagram of the $SnO_2$/$TiO_2$ heterojunction and photocatalysis mechanism.

## 3. Experimental Section

### 3.1. Chemical and Substrate

Pure tin oxide (SnO) (0.2 g, 99.995% pure, LTS Research Laboratories, Inc., USA) was used as the thermal evaporation material. The titanium tetra-isopropoxide (TTIP) precursor for ALD was purchased from UP Chemical Co., Ltd., Pyeongtaek, Korea. The porous SnO films were fabricated on Si (100) wafers or ITO substrates with the dimensions of $2 \times 2$ cm$^2$. The substrate was cleaned thoroughly with deionized water, acetone, and ethanol for 10 min and were dried by blowing $N_2$ over the sample. The UV-vis absorption spectra were measured using a $SnO_2$/$TiO_2$ photocatalyst on an indium-doped tin oxide (ITO) substrate (7–10 ohm/sq, transmittance: 80–85%).

### 3.2. Preparation of Porous $SnO_2$ Nanoform

The highly porous SnO nanoform was fabricated by Ar-assisted modified thermal evaporation, as previously reported [33,34]. The SnO source was kept in a Ta boat, which was subsequently placed at the center of the chamber, and the Si/SiO$_2$ substrate was placed at the upper site, with the cooling system at 23 °C. During thermal evaporation, the highly pure Ar gas (99.9999%) flowed with 100 sccm to reach a pressure of 0.2 Torr after evacuation to a $5 \times 10^{-6}$ Torr pressure, and the SnO source was rotated at a speed of 8 rpm in order to form a uniform nanoporous structure. After thermal deposition, the porous SnO nanoform was annealed at 700 °C for 1 h with 50 sccm pure air in a tube furnace to obtain the $SnO_2$ crystallite structure.

### 3.3. Preparation of the Porous $SnO_2$/$TiO_2$ Heterostructure

The highly porous $SnO_2$/$TiO_2$-10, 25, 50 and 100 heterostructure films was synthesized by atomic layer deposition with a different number of cycles on the porous $SnO_2$ nanofoam. The TTIP and deionized water were used as the metal precursor and oxygen source, respectively. Highly pure $N_2$ gas (99.9999%) was used for the purge process with a gas flow rate of 200 sccm. The as-synthesized porous $SnO_2$ film was kept at 300 °C in a reaction chamber, and the base pressure was maintained at $1.5 \times 10^{-2}$ torr using a dry pump (Edwards, IQDP80, Edwards. Co., United Kingdom). In order to control the uniformity and thickness of the $TiO_2$ layer, it was coated with a highly porous $SnO_2$ nanocomposite with 10, 25, 50, and 100 cycles of ALD, denoted as $SnO_2$/$TiO_2$-10, $SnO_2$/$TiO_2$-25, $SnO_2$/$TiO_2$-50, and $SnO_2$/$TiO_2$-100, respectively.

### 3.4. Evaluation of Photocatalytic Activity

The photocatalytic properties of the as-synthesized $SnO_2/TiO_2$-10, 25, 50, and 100 films were evaluated by the degradation of MB under UV light irradiation (265 nm) in a cooling bath. MB (1.2 mg) was dissolved in 1 L of deionized water and was stirred for 2 h in the dark. Before evaluating the photocatalytic properties, the porous $SnO_2/TiO_2$ heterostructured films were kept in 50 mL of MB solution and were stirred for 30 min in the dark to achieve adsorption–desorption equilibrium. After UV irradiation with a power of 16 W, the solution was characterized by UV-vis absorbance spectroscopy (UV-2500 UV-vis spectrophotometer, Shimadzu. Co., Japan) to evaluate the photodegradation of MB.

### 3.5. Materials Characterization

The microstructure of the $SnO_2/TiO_2$ nanoform was characterized by field-emission scanning electron microscopy (FE-SEM, JEOL JSM-7100F, JEOL Co., Japan) with energy dispersive X-ray spectroscopy (EDS) mapping. Elemental mapping analysis of $SnO_2/TiO_2$ from electron energy loss spectroscopy (EELS) were conducted using a ZEISS Libra 200 HT Mc Cs, Zeiss, Co., Germany (200 kV)/JEOL ARM 200 F, JEOL. Co., Japan (200 kV). The chemical binding energy of the $SnO_2/TiO_2$ nanoform with the number of $TiO_2$ deposition cycles was analyzed through X-ray photoelectron spectroscopy (XPS) using a K-alpha plus (ESCALAB 250Xi, Thermo Fisher Scientific. Co., USA) instrument utilizing monochromatic Al K$\alpha$ (1486.6 eV) radiation. The surface composition of Ti in $SnO_2/TiO_2$ was determined using a Rigaku ZSX Primus II energy-dispersive X-ray fluorescence (EDXRF) spectrometer (Rigaku. Co., Japan).

## 4. Conclusions

A highly porous $SnO_2/TiO_2$ heterojunction structure was synthesized using gas flow thermal evaporation and ALD, which controlled the $TiO_2$ thickness on the porous $SnO_2$ structure. Owing to the charge separation that leads to the formation of radicals, the heterojunction structure with a $TiO_2$ layer on $SnO_2$ exhibited a higher photocatalytic efficiency than bare $SnO_2$. The effect of $TiO_2$ thickness on the photocatalytic activity was also investigated. According to the results, $SnO_2/TiO_2$-50, which was analyzed to have $TiO_2$ deposited sparsely on the porous $SnO_2$, had the highest kinetic constant of 0.013 min$^{-1}$, which was two times higher than that of the core-shell structure of $SnO_2/TiO_2$-100. The facile separation of the electron/hole pairs and corresponding oxygen radicals on the surface of both $TiO_2$ and $SnO_2$ can improve the photocatalytic activity in the heterojunction compared to the core-shell structure.

**Supplementary Materials:** The following are available online at https://www.mdpi.com/article/10.3390/catal11101144/s1, Figure S1: EDX mapping for $SnO_2/TiO_2$-50: (a) lateral SEM image of $SnO_2/TiO_2$-50 of information collection area, (b) O element, (c) Si element (d) Sn element, and (e) Ti element, Figure S2: Growth per cycle (GPC) of $TiO_2$ thin film at 300 °C on Si substrate, Figure S3: EELS spectrum of $SnO_2/TiO_2$-50. Sn and Ti clearly reveal presence as heterojunction structure, Figure S4: XRD pattern of as-deposited $TiO_2$ layer by ALD cycles process on Si substrate, Figure S5: UV-Visible absorbance spectra of photodegradation of methylene blue (a–e) by $SnO_2$ and all $SnO_2/TiO_2$ with different ALD cycles, Figure S6: The photocatalytic efficiency (%) with increasing degradation time by $SnO_2$ and all $SnO_2/TiO_2$-50 photocatalysts.

**Author Contributions:** Conceptualization, S.K. and Y.M.P.; methodology, S.K., H.-K.C. and H.-J.K.; validation, S.K., K.B.K. and H.-N.L.; formal analysis, S.K.; investigation, S.K., K.B.K., T.J.P. and Y.M.P.; data curation, S.K.; writing—original draft preparation, S.K. and Y.M.P.; writing—review and editing, S.K. and Y.M.P.; project administration, Y.M.P.; All authors have read and agreed to the published version of the manuscript.

**Funding:** This research was financially supported by the Basic Research Program of the Korea National Research Foundation (Project No. NRF-2020R1F1A1067830).

**Conflicts of Interest:** The authors declare no conflict of interest.

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
