# Peer review of "Highly Porous SnO2/TiO2 Heterojunction Thin-Film Photocatalyst Using Gas-Flow Thermal Evaporation and Atomic Layer Deposition"

_catalysts, doi:10.3390/catal11101144_

Round 1
Reviewer 1 Report
The manuscript catalysts-1380200 entitled “Highly porous SnO2/TiO2 heterojunction thin-film photocatalyst using gas-flow thermal evaporation and atomic layer deposition” deals with the synthesis of TiO2 by atomic layer deposition onto SnO2 materials to be employed as efficient photocatalysts. A characterization of as synthesized SnO2 and the influence of TiO2 layer depositions on their chemical and optical properties are studied. Finally, the photocatalytic activity of the synthesized materials is demonstrated through the performance of photocatalytic assays for methylene blue (MB) degradation in water in presence of UV-C light and catalysts.
I consider that the hereby work need to improve the discussion of the results with some additional experimental tests and characterization analysis. Then, I would recommend revising the manuscript according to the comments listed above.
- The experimental section must be checked since the details of the characterization of SnO2/TiO2 materials are repeated in section 3.3. and 3.5. As consequence, the preparation of photocatalysts is not specified and it should be included in section 3.3.
- Regarding the photocatalytic experiments, how is the catalysts in the solution (e.g. in powered form, supported in a film)?
- According to lines 91-93, the changes on the porosity of SnO2 and coupled SnO2/TiO2 materials are shown by the FE-SEM analyses. However, the results of porosity of the SnO2/TiO2 could be better understood with the N2 adsorption-desorption analysis instead of SEM images. Thus, the value of surface area of both materials could be also better compared.
- According to Figure 3, the atomic Ti density is increased with the number of ALD cycles. Notwithstanding, no clear LINEAR trend is shown.
- As demonstrated with the XPS results, the deposition of TiO2 layers on the surface of the semiconductor SnO2, led to the weaker binding of oxygen atom due to the lower binding energy of TiO2 at 529.5 eV, compared to the binding energy of bare SnO2 at 530.4 eV. Moreover, a contribution of oxygen vacancies in the surface (531.2 eV) may be considered. Therefore, why the XPS spectrum of O1s is deconvoluted into two peaks in Figure 5? Could be calculated the proportion of each oxygen bond (Ti-O, Sn-O and Vo)?
- The band gap values cannot be corroborated caused of fact that the units of y-axis are missing. Moreover, if possible, the obtained values could be supported and compared with other studies.
- If possible, the full spectrum of MB adsorption and the UV-C lamp emission could be also provided (i.e. 200-800 nm). Thus, it could be given an idea of the degradation of MB by photolysis. Anyway, the result of this experiment (i.e. the photodegradation of MB) must be included in order to compare properly the efficiency of the photocatalytic process in presence of catalysts. Results could be included in Figure 7 and Table 1).
- Please, revise the legend of Figure S5(a).
- The use of photolysis in lines 184 and 186 is a bit confusing. Probably the term of photocatalysis is more appropriate for the experiments of the degradation of MB when light and catalyst are combined.
- Authors elucidate that SnO2/TiO2-100 exhibited less photocatalytic efficiency due to the fact that the superoxide radical is produced in a lower proportion than in SnO2/TiO2-50. However, this fact could be clearly demonstrated carrying out additional experiments with scavengers of hydroxyl radical, superoxide radical, holes and electrons.
- Is there a leakage of Ti or Sn in soultion?
In addition, some recommendations about the writing style could be taken into account:
- Use of personal style in the introduction should be avoided (i.e., line 78 “we also evaluated…”, line 81 “our results…”). I think that the use of an impersonal style is preferred.
- The sign of punctuation after the references of name of figures (e.g. [1-3]. Instead of . [1-3], or “. Figure X)” instead of “Figure X.”). Sometimes is not clear if references of Figures are related to the previous or the following sentence.
Author Response
Please see the attachment.
The file is the response to the reviewer's comment including the changed manuscript.

Reviewer 2 Report
- This paper is well written in a very concise and easy-understanding fashion. The effect of layer thickness of TiO2 on SnO2 is well illustrated. The explanation is reasonable.
- One issue authors should be aware of is the blue shift and red shift mentioned in the results and discussion section when discussing about the layer thickness on optical band gap. Be aware that the blue shift is the shift toward shorter wavelength, thus is toward larger band gap. Please check.
- The sentence “The Lo states from the oxygen atoms corresponding to fully coordinated SnO2 were gradually decreased with increasing TiO2 ALD cycles. (Figure 5)” cannot be verified directly from Figure 5, since the Lo states from the oxygen atoms corresponding to fully coordinated SnO2 were not been deconvoluted from that of TiO2. It can only be interpreted from the peak shift. Please verify.
4. The sentence “The photocatalytic efficiency of the porous SnO2 films is attributed to the lower conduction band of the SnO2 compared to the O2/∙O2- oxidation potential,…” may be misleading when compare electron energy with electric potential. I guess authors would like to express “The photocatalytic efficiency of the porous SnO2 films is attributed to conduction band of the SnO2, which has lower electric potential compared to the O2/∙O2- oxidation potential,…”. Please verify.
Author Response

(The authors gave the same response as above.)

Round 2
Reviewer 1 Report
*The values of "axis y" on Figure 6b are missing.
* Please, revise the punctuation marks. The period must be at the end of the sentence (eg. line 99: "...surface of the porous SnO2 by ALD. (Figure 2.)" should be "...surface of the porous SnO2 by ALD (Figure 2).")
Author Response
Thanks for kind indication. As to reviewer's comment, we modified the position of period for entire manuscript. Regarding to the value of y-axis in Figure 6(b), it is calculated result from figure 6(b), so it is also arbitrary unit. We also corrected it as the reviewer suggest.
